# Human-like multiple object tracking through occlusion via gaze-following

**Benjamin Peters**[*]                                    BENJAMIN.PETERS@GLASGOW.AC.UK
*School of Psychology and Neuroscience, University of Glasgow, UK*

**Eivinas Butkus**[*]                                    EIVINAS.BUTKUS@COLUMBIA.EDU
*Department of Psychology, Columbia University, New York, NY 10027*

**Nikolaus Kriegeskorte**                                N.KRIEGESKORTE@COLUMBIA.EDU
*Zuckerman Mind Brain Behavior Institute and Departments of Psychology, Neuroscience, and Electrical Engineering, Columbia University, New York, NY 10027*

## Abstract

State-of-the-art multiple object tracking (MOT) models have recently been shown to behave in qualitatively different ways from human observers. They exhibit superhuman performance for large numbers of targets and subhuman performance when targets disappear behind occluders. Here we investigate whether human gaze behavior can help explain differences in human and model behavior. Human subjects watched scenes with objects of various appearances. They tracked a designated subset of the objects, which moved continuously and frequently disappeared behind static black-bar occluders, reporting the designated objects at the end of each trial. We measured eye movements during tracking and tracking accuracy. We found that human gaze behavior is clearly guided by task relevance: designated objects were preferentially fixated. We compared human performance to that of cognitive models inspired by state-of-the-art MOT models with object slots, where each slot represents the model's probabilistic belief about the location and appearance of one object. In our model, incoming observations are unambiguously assigned to slots using the Hungarian algorithm. Locations are tracked probabilistically (given the hard assignment) with one Kalman filter per slot. We equipped the computational models with a fovea, yielding high-precision observations at the center and low-precision observations in the periphery. We found that constraining models to follow the same gaze behavior as humans (imposing the human-measured fixation sequences) best captures human behavioral phenomena. These results demonstrate the importance of gaze behavior, allowing the human visual system to optimally use its limited resources.

**Keywords:** human dynamic object vision, multiple object tracking, visual occlusion, eye movements, computational modelling

## 1. Introduction

Human vision parses the world into representations of objects. These representations persist as objects in the world move, leave our field of view or disappear behind occluding objects. This ability emancipates vision from the immediate sensory input and enables us to see the world in terms of its physical constituent components, providing a basis for prediction and successful action. Such robust and efficient inference of task-relevant objects from the

---

[*] shared first authors

sensorium is a challenge for machine vision and so far lacks clear solutions (Peters and Kriegeskorte, 2021; Greff et al., 2020).

Human vision may, in part, derive its efficiency and robustness from targeted sampling of the environment through foveation (Gegenfurtner, 2016). Understanding inference-driven human foveation, i.e., why people look where, may inspire novel robust and data-efficient machine vision models. Understanding (and thus being able to predict) human foveation (Kümmerer and Bethge, 2023) is also of high interest for applications like computer graphics (Padmanaban et al., 2017), accessibility (Ward and MacKay, 2002), UX design (Bylinskii et al., 2017), and relevant for the diagnostics of neurological and psychiatric disorders (Liu et al., 2021).

Multiple object tracking (MOT) is a class of tasks that directly taps into a system's ability to stably represent task-relevant objects in a dynamic visual input. MOT tasks have a long tradition in cognitive science (Pylyshyn and Storm, 1988; Scholl and Pylyshyn, 1999) and machine learning. We recently compared state-of-the art MOT models (Bewley et al., 2016; Wojke et al., 2017; Zhang et al., 2022; Cao et al., 2022) to humans on a novel task designed to take steps toward bridging the gap between the real-world complexity of machine learning tasks and the abstraction of cognitive tasks (Peters et al., 2022). The task combined object recognition demands, visual occlusions and tracking. We observed that models displayed qualitatively different behavior from humans: their performance was independent of the set size, i.e. the number of objects, but deteriorated below human performance when objects were subject to extended periods of full occlusions.

This study investigates whether the qualitative difference in behavior between models and humans may be due to the fundamental differences in how models and humans sample the world. In contrast to standard machine vision models, human vision obtains high-precision observations at the fovea and low-precision observations in the periphery, actively sampling the environment to build a visual representation of the world that is useful for current behavioral goals. Human gaze behavior has been shown to be highly relevant for successful object tracking (Fehd and Seiffert, 2008; Zelinsky and Neider, 2008; Hyönä et al., 2019) and individual gaze behavior is predictive of MOT task performance (Upadhyayula and Flombaum, 2020).

We here equip a computational model, inspired by machine learning multiple object tracking models, with two features of human vision. We first introduce observation noise to the model, lowering the fidelity of the visual input. We then equip the model with a fovea and constrain the model to follow the gaze trajectories of humans performing the same trial. We find that both features lead to more human-like performance as a function of set size and occlusion.

## 2. Related work

### 2.1. Cognitive science

Previous work in cognitive science has modeled human multiple object tracking behavior in highly abstracted tasks, where objects have the same appearance (i.e., dots without texture) and are visible throughout the motion period (i.e., no occlusion) (Zhong et al., 2014; Vul et al., 2009; Srivastava and Vul, 2016). E.g., Vul et al. (2009) used a probabilistic slot-model (i.e., particle filter) to explain human multiple-object tracking behavior as a function of the

number of objects, velocity, and object distance. Srivastava and Vul (2016) incorporated an attentional controller into the particle filter that modulates the attentional gain of internal object representations, explaining moment-by-moment spatial precision of tracked object representations. Here, we use gaze behavior to align the beliefs of computational models with those of human observers. Similar to Upadhyayula and Flombaum (2020), who could explain individual differences in MOT performance by "following" the gaze of individual participants, we implement gaze-following by incorporating fixation distance-dependent noise. Our work goes beyond the highly abstracted tasks typically employed in cognitive science, incorporating object appearance information, and modeling how active sampling of positions and appearances of objects may help maintain object representations through extended periods of occlusion.

## 2.2. Machine learning

A central area of machine vision research is the development of real-time online methods to, e.g., track multiple objects in videos (Luo et al., 2021; Bewley et al., 2016; Wang et al., 2020). This may require online mechanisms that selectively process and combine certain aspects of the available information. Such attention mechanisms have been employed in computer vision (Guo et al., 2022; Sun et al., 2021; Meinhardt et al., 2022; Zeng et al., 2022), including spatial attention mechanisms (akin to overt and covert attention in humans) (Larochelle and Hinton, 2010; Mnih et al., 2014; Eslami et al., 2016) and in tracking tasks (Denil et al., 2012; Kahou et al., 2017; Kosiorek et al., 2017, 2018). While advancements in hardware and architectures (Vaswani et al., 2017) may eventually admit online processing of the full input for standard fixed camera MOT benchmarks (Milan et al., 2016), spatial sampling will remain highly relevant in areas like active robot vision (Zeng et al., 2020).

## 3. Methods

### 3.1. Model

The computational model is a slot-based model, which tracks object identities over time by maintaining and updating beliefs about the position and appearances of objects in the scene (Fig. 1). The model is inspired by modern multiple object tracking models from machine learning like SORT (Bewley et al., 2016) and DeepSORT (Wojke et al., 2017). SORT is a simple and fast algorithm that for each of $N$ objects, tracks a Gaussian belief about the object's location and size (represented as a bounding box) and its velocity. On each frame, $M$ new observations are detected via an object detection network. These observations are then assigned to the current tracks (slots) such that the overall distances between observations and tracks is minimized by the assignments (using the Hungarian algorithm). Object beliefs are then updated using Kalman filtering. DeepSORT extends SORT with a deep appearance-based association metric. On each frame, a crop is extracted for each observation from the detected bounding box and embedded into latent space that was optimized to disambiguate tracked objects by their appearances. Each slot then, in addition to the Gaussian belief about the position, keeps a library of embeddings that have been previously associated with the track. The assignment of observations to tracks is then

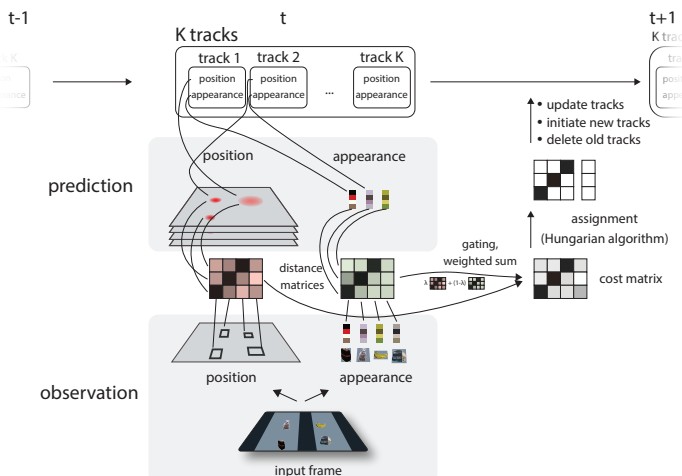

Figure 1: **Tracking model.** The base tracking model is a slot-based tracking-by-detection model. See text for details.

obtained by minimizing a combined cost of position-related beliefs and observations, as well as the distance between observed embeddings and the memory of past embeddings.

### 3.1.1. Base model

On the first frame, the model initializes a series of tracks (slots), one for each detection, representing the objects to be tracked. Each track represents the belief about an object's position and appearance. In particular, the $i$-th object's position state at time $t$ is an eight-dimensional variable (bounding box center, height, aspect ratio, and their velocities) and the belief over this variable is represented as Gaussian distribution with mean $\mathbf{z}_{i,t}^{\text{pos}}$ and covariance $\mathbf{P}_{i,t}^{\text{pos}}$. Similarly, the object's appearance is represented as a Gaussian belief in a 128-dimensional deep embedding space. Mean $\mathbf{z}_{i,t}^{\text{app}}$ and covariance $\mathbf{P}_{i,t}^{\text{app}}$ are the empirical moments computed over the past 10 embeddings associated with track $i$ at time step $t$.

Assignment of tracks to new observations is performed by minimizing an assignment cost using the Hungarian algorithm. We use the negative log-likelihood of position and appearance observations under the tracked beliefs about position and appearance. For position, we project the Gaussian filter belief about the track $i$ into the observation space forming the prediction $\mathcal{N}(\hat{\mathbf{o}}_{i,t}^{\text{pos}}, \mathbf{S}_{i,t}^{\text{pos}})$. In the base model (without observation noise), the uncertainty about the object's location in observation space $\mathbf{S}_{i,t}^{\text{pos}}$ is purely a function of the belief uncertainty: $\mathbf{S}_{i,t}^{\text{pos}} = \mathbf{H}\hat{\mathbf{P}}_{i,t}^{\text{pos}}\mathbf{H}^{\mathsf{T}}$. Similarly, in the base model, the prediction of appearance is purely a function of the belief uncertainty about appearance $\mathcal{N}(\mathbf{z}_{i,t}^{\text{app}}, \mathbf{P}_{i,t}^{\text{app}})$.

The position and appearance costs, $d^{\text{pos}}(i, j)$ and $d^{\text{app}}(i, j)$ (see Appendix A for details), of associating track $i$ with detection $j$ are then the negative log-likelihoods of observed position and appearance of detection $j$ that are combined into a single cost $c(i, j)$ using $\lambda$:

$$c(i, j) = (1 - \lambda)d^{\text{pos}}(i, j) + \lambda d^{\text{app}}(i, j) \tag{1}$$

We set $\lambda = 0.056$ to roughly match the negative log-likelihood cost scales for position and appearance, but we did not fit it to human behavior. After an observation is assigned to a track, the track updates its belief about the position using the Kalman update and incorporates the new embeddings into its embedding library.

### 3.1.2. OBSERVATION NOISE MODEL

The observation noise model adds constant normally distributed noise to the position detections $\tilde{\mathbf{o}}_{j,t}^{\text{pos}}$ and incoming embeddings $\tilde{\mathbf{o}}_{j,t}^{\text{app}}$:

$$\mathbf{o}_{j,t}^{\text{pos}} = \tilde{\mathbf{o}}_{j,t}^{\text{pos}} + \mathcal{N}(\mathbf{0}, \frac{\sigma_{\text{pos}}^2}{\tau_{j,t,\text{pos}}^2}\mathbf{I}) \tag{2}$$

$$\mathbf{o}_{j,t}^{\text{app}} = \tilde{\mathbf{o}}_{j,t}^{\text{app}} + \mathcal{N}(\mathbf{0}, \frac{\sigma_{\text{app}}^2}{\tau_{j,t,\text{app}}^2}\mathbf{\Sigma}^{\text{app}}) \tag{3}$$

Here, $\tilde{\mathbf{o}}_{j,t}^{\text{pos}}$ is the ground truth location of detection $j$ (which is known to us because we control the generation of the stimuli), while $\tilde{\mathbf{o}}_{j,t}^{\text{app}}$ is the embedding associated with detection $j$ obtained by passing the bounding-box-cropped image to a neural network that extracts visual features. $\mathbf{\Sigma}^{\text{app}}$ denotes the empirically estimated covariance of the embeddings, and it is used to "color" the noise of the embeddings according to the covariance structure of the embedding space. We simulated model behavior with $\sigma_{\text{pos}} = [15.0, 25.0]$ and $\sigma_{\text{app}} = [0.2, 0.3]$ as these approximately performed at human accuracy levels. When comparing to human behavior, we averaged across these noise levels, as they had negligible effects on the results. Since we treat the observation noise model as a special case of the fixation model, the precision values $\tau_{j,t,\text{pos}}^2$ and $\tau_{j,t,\text{app}}^2$ are are fixed at 22.5 (which is the inverse of the intercept $c$ from the cortical magnification formula below).

### 3.1.3. FIXATION MODEL

Finally, we implemented an artificial fovea that spatially modulates the observation noise. Following neuroscientific literature (Dumoulin and Wandell, 2008; Rovamo and Virsu, 1979; Carrasco et al., 1995), detections in the fovea have a lower observation noise than detections in the periphery. This modification allows us to feed human eye fixation data to the model, and make the model "fixate" according to human gaze.

In practice, for each detection, we calculate its eccentricity (distance from fixation) $E$ in visual angles. We follow vision science literature (Strasburger et al., 2011) and find the precision for appearance and position using an inverse linear function that relates eccentricity $E$ to cortical magnification:

$$\tau_{j,t,\text{pos}}^2 = 1/(b_{\text{pos}} \times E + c) \times n_{\text{pos}} \tag{4}$$

$$\tau_{j,t,\text{app}}^2 = 1/(b_{\text{app}} \times E + c) \times n_{\text{app}} \tag{5}$$

Slopes for position and appearance precision $b_{\text{app}}$ and $b_{\text{pos}}$ are parameters in our model, while $c = 0.044$ is fixed according to plausible values from human peripheral vision literature (Strasburger et al., 2011). Intuitively, the slope parameters determine how fast the precision falls off with distance from fixation. We can recover the constant observation model by setting the slopes to 0, rendering precision independent of eccentricity $E$. The normalization factors $n_{\text{pos}}$ and $n_{\text{app}}$ ensure that average precision across the screen is the same for different slope values.

Importantly, the fixation model expects higher levels of observation noise for tracks that are far away from fixation. So it computes *track-based* precision values for each track $i$ $\tau^2_{i,t,\text{pos}}$ and $\tau^2_{i,t,\text{app}}$ that are then used to make predictions for that track in position and embedding observation spaces (note that these are different from the detection-based precision values used in simulating observation noise in the detections). For instance, the uncertainty about the object's location in observation space $\mathbf{S}^{\text{pos}}_{i,t} = \mathbf{H}\hat{\mathbf{P}}^{\text{pos}}_{i,t}\mathbf{H}^{\mathsf{T}} + \mathbf{R}^{\text{pos}}_{i,t}$ becomes a function of the belief uncertainty $\hat{\mathbf{P}}^{\text{pos}}_{i,t}$ and fixation-dependent observation noise $\mathbf{R}^{\text{pos}}_{i,t} = \frac{\sigma^2_{\text{pos}}}{\tau^2_{i,t,\text{pos}}}\mathbf{I}$.

### 3.2. Task

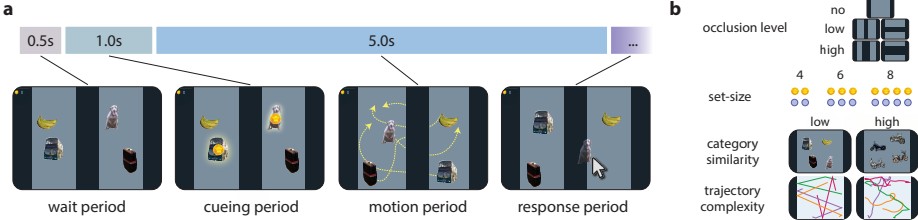

Figure 2: **Task**. **a** Sequence of events. **b** Experimental factors (see 3.2 for details).

The object-tracking task was identical for both humans and models, unless indicated differently (see Figure 2a). Each trial started with the display of either 4, 6, or 8 objects. In Phase 1, the *cueing period*, objects remained still for 1.5s. During this time, half of the objects were highlighted as target objects for human participants for 1s. Phase 2, the *motion period*, lasted five seconds, where objects moved independently on random trajectories inside the motion area. Objects moved with constant speed, and their trajectory was simulated either through a linear or more complex non-linear dynamics model (see Appendix B for details). Object speed vectors were reflected at the boundaries of the motion area. Two-thirds of all trials contained a central large rectangle ("occluder") that spanned the full vertical (horizontal) extent and either 20% or 40% of the horizontal (vertical) extent of the motion area. Object starting positions were sampled such that (1) all objects were unoccluded at the start of the trial and (2) all objects ended up in positions such that no two were closer to each other than a threshold distance. In phase 3, the *response period*, the objects stopped moving and target identification was required. Humans did so by clicking on them. If objects ended their motion hidden by the occluder, they were made visible either by making the occluder semi-transparent (for humans), or by moving their depth plane in front of the occluder (for models, which were not assumed to be familiar with semi-

transparency). Object stimuli were crops extracted from the MS Coco challenge (Lin et al., 2014) using the provided segmentation masks. For each of the 80 categories, we extracted a large set of different exemplars, excluding fragmented objects.

Humans and models were presented with 144 motion sequences, for which we varied several factors (Figure 2b). We varied the number of objects (4, 6, or 8; always half of them were targets), the extent of the occluder (no occluder, 20%, 40%), and its orientation (horizontal or vertical). Object dynamics were either linear or following a more complex motion model. Object appearances could be sampled either randomly from all 80 Coco categories or only from a single category.

### 3.3. Human behavior

For the human participants ($N = 9$), target objects were highlighted in the cueing phase with a glowing outline and a coin icon placed atop each target. In the response phase, participants selected half of the objects which they believed were the original targets. Once all responses were made, the selected objects' identities were revealed (target or non-target) by displaying an animation of the coin being collected for targets and no animation for selected non-targets. Humans were not constrained by a time limit for their responses. Trials were presented in 12 blocks of 12 sequences (trials) each. Human gaze position was recorded simultaneously using an eyetracker (see Appendix C).

### 3.4. Model behavior

Following previous work (Peters et al., 2022), we provide models with ground truth bounding boxes for unoccluded objects in the current frame. This is similar to public detections in the MOT challenge (Milan et al., 2016) allowed us to focus on the tracking challenge rather evaluating the quality of the object detector. To obtain responses to objects at the end of the motion period, we determined whether a model tracked the target objects consistently throughout the whole trial. A target object was considered to be successfully tracked if its final detection was assigned to one of the target tracks of the first frame. The model then "selected" those final detections that were assigned to a target track. In case less than $T$ detections were assigned to target tracks, the model randomly selected responses from the remaining detections (i.e., guessing) until $T$ responses were made. We report the average accuracy over $2 \times 9 = 18$ model evaluation runs (2 independent runs for each of the 9 participant gaze data, or simply 18 runs in the case of no noise or constant noise models).

## 4. Results

### 4.1. Human behavioral phenomena

We assessed the impact of the number of objects, occlusion levels, object category similarity, and trajectory complexity (see Figure 3). We found that tracking performance decreases with the number of objects tracked and the degree of occlusion, replicating previous findings in the MOT literature (Intriligator and Cavanagh, 2001; Pylyshyn and Storm, 1988; Scholl and Pylyshyn, 1999; Yantis, 1992). We also found that performance diminishes when targets and distractors belong to the same category, replicating our own previous findings (Peters et al., 2022). Moreover, tracking performance was slightly reduced when objects moved on

more complex trajectories compared to linear trajectories. We were particularly interested in capturing these broad patterns of human object-tracking behavior with our computational modeling efforts.

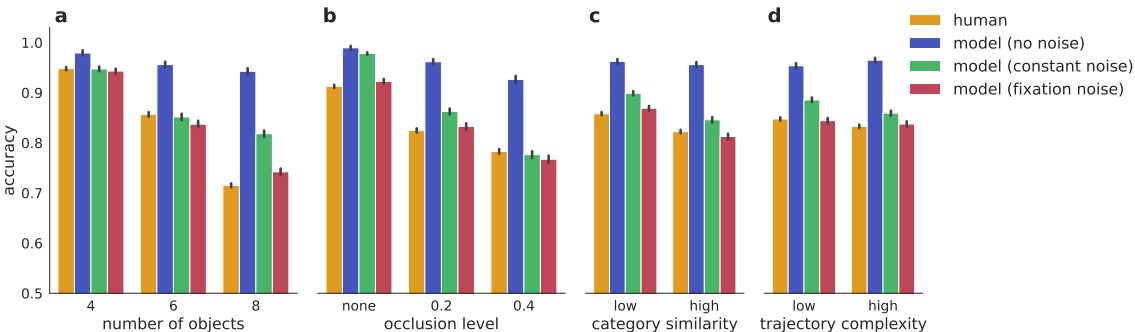

Figure 3: **Human and model accuracy. a.** Number of objects (half of them were targets), **b** Occlusion level (no occlusion, 20%, and 40% width of the occluder). Low and high category similarity (**c**) and trajectory complexity (**d**).

## 4.2. Models without fixation

We first modeled human behavior without any observation noise in the model, setting $\sigma_{pos} = 0$ and $\sigma_{app} = 0$. As seen in Figure 3 (blue bars, "model (no noise)"), this model outperforms humans in all conditions, yielding a qualitatively bad fit to human data.

In contrast to this model, humans have limited precision in their observations. In the next step, we therefore added observation noise to all detections (observed positions and appearance embeddings, as explained in the Methods section). This observation noise was identical (constant) across the whole visual field. We find that adding constant observation noise made the model capture human phenomena much more closely than the model without observation noise (Figure 3, green bars, "model (constant noise)"). However, state-of-the-art object tracking models tend to outperform humans when there is no occlusion and underperform when there is high occlusion (Peters et al., 2022). The constant noise model seems to still suffer from this problem (see Figure 3B).

## 4.3. Introducing gaze-following makes model behavior more human-like

We hypothesized that a model equipped with a fovea and following human gaze would yield a better fit to human behavior. In particular, we observed that human gaze behavior is directly related to the task, revealing a participant's beliefs about tracked objects (see Appendix D). A qualitative inspection of the results (Figure 3, red bars, "model (fixation noise)") suggests that the fixation model does indeed better capture human behavior than the constant noise model. In particular, unlike the constant noise model, the fixation model does not seem to outperform humans in the lower occlusion settings ("None" and "0.2" in Figure 3B). To quantify the effects of introducing fixations into the model, we computed

the correlation between model and human average accuracy across the 36 experimental conditions (3 levels of the number of objects × 3 occlusion levels × 2 levels of category similarity × 2 levels of trajectory complexity) (Figure 4A). This metric reflects the extent to which the model captures key human behavioral phenomena while being invariant to overall level of performance (Figure 3).

We found that the best-performing fixation model ($b_{\mathrm{pos}} = 1.0, b_{\mathrm{app}} = 0.055$) captured human phenomena significantly better than a model without fixation (i.e., $b_{\mathrm{pos}} = 0, b_{\mathrm{app}} = 0$) ($t(8) = 8.29$, $p < .001$). In fact, every fixation model with non-zero slope for both fixation and appearance is a better fit to human behavior than any model in which either the appearance or the position slope is set to 0 (for any comparison, $t(8) > 2.33$, $p < 0.048$); and every model with one non-zero slope is better than the zero-slope model (for any comparison, $t(8) > 3.13$, $p < 0.014$). See Figure 4B first column/row for a visual intuition. In general, we found that model correlation to human behavior increases for higher slopes (Figure 4C), suggesting that fixations are indeed key to explaining human tracking behavior.

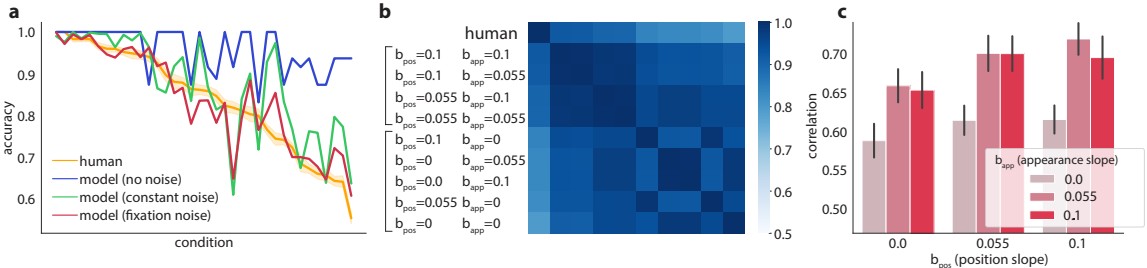

Figure 4: **Human and model fit. a** Accuracy of humans and models for each experimental condition (sorted by human accuracy in each condition). **b** Similarity between human and model behavior across experimental conditions. Note that models with at least one 0-slope (no fixation effect for observed position, appearance, or both, i.e., the last five rows/columns of the matrix) have a lower correlation to human behavior. **c** Correlation to human behavior as a function of position slope $b_{\mathrm{pos}}$ and appearance slope $b_{\mathrm{app}}$. Models behave more human-like when increasing the slope (i.e. when increasing the effect of fixation).

## 5. Limitations and future work

Our model is a slot-based tracking-by-detection algorithm, which allowed us to isolate the tracking challenge from the detection components of the task. Tracking-by-detection is a well-established paradigm in machine vision (Bewley et al., 2016; Wojke et al., 2017). However, the separation of detection from association is an oversimplification of human vision, in which detection and association across time and space occur continuously and concurrently at all stages of visual processing. Departing from the classic tracking-by-detection paradigm (e.g., Sun et al., 2021; Feichtenhofer et al., 2017; Bergmann et al., 2019) and perhaps even including less structured (i.e., non-slot) representations of the visual

world (Eslami et al., 2018; Vondrick et al., 2018) may yield a closer alignment of human and machine vision.

## 6. Conclusion

We here equipped a computational model, inspired by state-of-the-art multiple object tracking models in machine learning, with a fovea, yielding high-precision observations at the center and low-precision observations in the periphery. We found that constraining models to follow the same gaze behavior as humans (imposing the human measured fixation sequences) better captures key behavioral human phenomena compared to models without fixation. These results demonstrate the importance of gaze behavior and provide a stepping stone for building resource-efficient machine vision models that sample their environment adaptively.

## 7. References

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

## Acknowledgments

B.P. has received funding from the EU Horizon 2020 research and innovation programme under the Marie Skłodowska-Curie grant agreement no. 841578. This work was also supported by the National Science Foundation under Grant No. 1948004 to N.K. We thank Qiyi Peng for help with the data acquisition.

## Appendix A. Model details

On the first frame, the tracker initializes a series of tracks, one for each detection, representing the objects to be tracked. Each track represents the belief about an object's position variables (bounding box center, height, aspect ratio, and velocities) and appearance. New observations are assigned deterministically to tracks/slots and update beliefs about position and appearance.

We denote observations with $\mathbf{o}$ (position observations: $\mathbf{o}^{\mathrm{pos}}$, appearance observations: $\mathbf{o}^{\mathrm{app}}$) and latent variables with $\mathbf{z}$ (position latent variables: $\mathbf{z}^{\mathrm{pos}}$, appearance latent variables: $\mathbf{z}^{\mathrm{app}}$).

### A.1. Position model

Each track's position state is an eight-dimensional variable (bounding box center, height, aspect ratio, and velocities), which is updated with new observations by a Kalman filter.

On time step $t$, first the predicted posterior $\mathcal{N}(\hat{\mathbf{z}}_{i,t}^{\mathrm{pos}}, \hat{\mathbf{P}}_t^{\mathrm{pos}})$ is computed with

$$\hat{\mathbf{z}}_{i,t}^{\mathrm{pos}} = \mathbf{F}\mathbf{z}_{i,t-1}^{\mathrm{pos}} \tag{6}$$

$$\hat{\mathbf{P}}_t^{\mathrm{pos}} = \mathbf{F}\mathbf{P}_{t-1}^{\mathrm{pos}}\mathbf{F}^{\mathsf{T}} + \mathbf{Q} \tag{7}$$

where $\mathbf{z}_{t-1}^{\mathrm{pos}}$, and $\mathbf{P}_{t-1}^{\mathrm{pos}}$ are the state vector and covariance of the posterior belief from the previous frame, $F$ is the state-transition matrix:

$$\mathbf{F} = \begin{bmatrix} 1 & 0 & 0 & 0 & \Delta t & 0 & 0 & 0 \\ 0 & 1 & 0 & 0 & 0 & \Delta t & 0 & 0 \\ 0 & 0 & 1 & 0 & 0 & 0 & \Delta t & 0 \\ 0 & 0 & 0 & 1 & 0 & 0 & 0 & \Delta t \\ 0 & 0 & 0 & 0 & 1 & 0 & 0 & 0 \\ 0 & 0 & 0 & 0 & 0 & 1 & 0 & 0 \\ 0 & 0 & 0 & 0 & 0 & 0 & 1 & 0 \\ 0 & 0 & 0 & 0 & 0 & 0 & 0 & 1 \end{bmatrix}$$

and the covariance of the process noise $\mathbf{Q} = \begin{bmatrix} 2.1, 2.1, 0.01, 2.1, 0.26, 0.26, 1 \times 10^{-5}, 0.26 \end{bmatrix}\mathbf{I}$.

The new posterior for time step $t$ is then updated as a combination of the predicted posterior and the assigned new observation $\mathbf{o}_{j,t}$, weighted by the Kalman gain:

$$\mathbf{z}_{i,t}^{\mathrm{pos}} = (\mathbf{I} - \mathbf{K}_t\mathbf{H})\hat{\mathbf{z}}_{i,t}^{\mathrm{pos}} + \mathbf{K}_t\mathbf{o}_{j,t} \tag{8}$$

$$\mathbf{P}_{i,t}^{\mathrm{pos}} = (\mathbf{I} - \mathbf{K}_t\mathbf{H})\hat{\mathbf{P}}_{i,t}^{\mathrm{pos}} \tag{9}$$

where $\mathbf{H}$ projects latent beliefs into the observation space:

$$\mathbf{H} = \begin{bmatrix} 1 & 0 & 0 & 0 & 0 & 0 & 0 & 0 \\ 0 & 1 & 0 & 0 & 0 & 0 & 0 & 0 \\ 0 & 0 & 1 & 0 & 0 & 0 & 0 & 0 \\ 0 & 0 & 0 & 1 & 0 & 0 & 0 & 0 \end{bmatrix}$$

The Kalman gain weighs the contribution of the predicted posterior belief and the current observation to the new state belief, depending on the uncertainty in the predicted posterior $\hat{\mathbf{P}}_{i,t}^{\text{pos}}$ and the uncertainty over the expected observation $\mathbf{S}_{i,t}^{\text{pos}} = \mathbf{H}\hat{\mathbf{P}}_{i,t}^{\text{pos}}\mathbf{H}^{\mathsf{T}} + \mathbf{R}_{j,t}^{\text{pos}}$, where $\mathbf{R}_{j,t}^{\text{pos}}$ is the covariance matrix of the observation noise. Note that observation noise is zero for the base model, constant ($\mathbf{R}_{j,t}^{\text{pos}} = \mathbf{R}^{\text{pos}}$) for the constant model, and a function of the distance between fixation and observation position in the fixation model.

## A.2. Appearance model

For each bounding box observation, the corresponding image crop is extracted and embedded into a latent space to yield a 128-dimensional appearance observation (extracted via a pre-trained re-identification model with a ResNet50 backbone). A track's belief about the object's appearance is modeled as an empirical distribution over past observations of the object. In particular, the Gaussian belief distribution $\mathcal{N}(\mathbf{z}_{i,t}^{\text{app}}, \mathbf{P}_{i,t}^{\text{app}})$ is parameterized via the precision-weighted mean $\mathbf{z}_t^{\text{app}}$ and covariance $\mathbf{P}_{i,t}^{\text{app}}$ over the past $K$ ($K = \min\{t, 10\}$) appearance embeddings. The precision weight of a sample in memory corresponds to the inverse variance of the observation noise associated with the observation.

The predicted appearance observation for a particular track $i$ at time-point $t$ is then modeled as $\mathcal{N}(\hat{\mathbf{o}}_{i,t}^{\text{app}}, \mathbf{S}_{i,t}^{\text{app}})$ with $\hat{\mathbf{o}}_{i,t}^{\text{app}} = \mathbf{z}_{i,t-1}^{\text{app}}$ and $\mathbf{S}_{i,t}^{\text{app}} = \mathbf{P}_{t-1}^{\text{app}} + \mathbf{R}_{j,t}^{\text{app}}$ (the projection $\mathbf{H}$ from latent into observation space is simply the identity matrix $\mathbf{I}$).

## A.3. Assignment

New observations are associated with those tracks that minimize the distances between the tracker's belief about object positions and appearances and the new observations.

In particular, we compute the distance $d^{\text{pos}}(i, j)$ between the tracker's belief about the $i$-th track's position and the position of the $j$-th observation $\mathbf{o}_{j,t}^{\text{pos}}$ as the negative log probability of the observation under the model's probabilistic prediction of the object's position, $\mathcal{N}(\hat{\mathbf{o}}_{i,t}^{\text{pos}}, \mathbf{S}_{i,t}^{\text{pos}})$, with $\hat{\mathbf{o}}_{i,t}^{\text{pos}} = \mathbf{H}\hat{\mathbf{z}}_{i,t}^{\text{pos}}$ as the predicted position in observation space.

$$d^{\text{pos}}(i, j) = \frac{1}{2}\left((\mathbf{o}_{j,t}^{\text{pos}} - \hat{\mathbf{o}}_{i,t}^{\text{pos}})^{\mathsf{T}}\left(\mathbf{S}_{i,t}^{\text{pos}}\right)^{-1}(\mathbf{o}_{i,t}^{\text{pos}} - \hat{\mathbf{o}}_{i,t}^{\text{pos}}) + \log|\mathbf{S}_{i,t}^{\text{pos}}| + k\log(2\pi)\right) \qquad (10)$$

Similarly, the distance between distance $d^{\text{app}}(i, j)$ between the tracker's belief about the $i$-th track's appearance and the observed appearance is computed as the negative log probability of the observed appearance embedding $\mathbf{o}_{j,t}^{\text{app}}$ under the model's belief about the track's appearance $\hat{\mathbf{o}}_{i,t}^{\text{app}}$ and the associated uncertainty $\mathbf{S}_{i,t}^{\text{app}}$

$$d^{\text{app}}(i, j) = \frac{1}{2}\left((\mathbf{o}_{j,t}^{\text{app}} - \hat{\mathbf{o}}_{i,t}^{\text{app}})^{\mathsf{T}}\left(\mathbf{S}_{i,t}^{\text{app}}\right)^{-1}(\mathbf{o}_{i,t}^{\text{app}} - \hat{\mathbf{o}}_{i,t}^{\text{app}}) + \log|\mathbf{S}_{i,t}^{\text{app}}| + k\log(2\pi)\right) \qquad (11)$$

on each time step, the entries of the assignment cost matrix $\mathbf{C}t$ is computed as a weighted sum of the position cost $D^{\text{pos}}$ and the appearance costs $D^{\text{app}}$ between the $i$-th track and the $j$-th observation.

$$c_{i,j} = \lambda d^{\text{pos}}(i, j) + (1 - \lambda)d^{\text{app}}(i, j) \qquad (12)$$

## Appendix B. Object motion trajectories

Initial object positions were sampled randomly such that no object was occluded and objects had a minimal inter-object distance of half an object width. Initial angular motion directions were sampled from a uniform distribution. Object speed was always constant. Using rejection sampling, motion trajectories were sampled such that at the end of the motion period, object centroids were separated by at least half an object distance.

Object motion dynamics could be either linear or following a complex generative motion model. In the linear case, object positions were deterministically simulated forward using the initial start position and motion vector. In the complex motion model, the angular motion direction was perturbed on each frame. At the start of the motion, the motion perturbation angle was sampled from a von Mises distribution centered on $\mu = 0$ degrees with a precision of $\kappa = 100.0$ degrees. This motion perturbation angle was then applied for the next $T_1$ frames. At frame $T_1 + 1$, a new motion perturbation angle was sampled from the same von Mises distribution and applied for the next $T_2$ frames. Intervals $T_1, T_2, \ldots$ were sampled from a Poisson distribution with $\lambda = 10$. Hence, on average every 10 frames, the motion direction of the object changed. This procedure yielded complex but smooth motion trajectories.

## Appendix C. Human gaze tracking

Nine participants (7 female, mean $\pm$ age $26.9 \pm 8.3$) with normal or corrected-to-normal vision were recruited from the participant pool of the Institute of Neuroscience and Psychology, University of Glasgow. All participants gave informed consent and the study was approved by the ethics committee of the College of Medical, Veterinary & Life Sciences of the University of Glasgow. Participants viewed stimuli in the lab on a monitor ($1920 \times 1080$ resolution, 60Hz refresh rate). Monocular gaze (left eye) was recorded at a sampling rate of 1000Hz using an infrared camera (Eyelink 1000, SR Research). The camera was positioned under the display monitor facing the participants. Participants used a chin-rest, which allowed us to control the distance between the eyes and the monitor (distance: 57cm) and minimized head motion. The eye tracker was calibrated before each block.

## Appendix D. Human gaze behavior reveals beliefs about tracked objects

Fixation behavior is a core feature of human visual inference enabling targeted sampling of the environment. To gauge to what extent fixations are directly subserving the task rather than being a behavior coincidental to the task, we plotted the distance of all objects to the fixation center as a function of time in the trial and object type (Figure 5). We observe that fixations were closer to targets compared to distractor objects in line with previous findings (Hyönä et al., 2019). Moreover, fixation behavior revealed the underlying beliefs of participants about target and distractor objects. In particular, distractor objects which were (incorrectly) selected by participants in the responses period, were closer to the fixation during the motion period compared to objects which were not selected. This demonstrates the relevance of human fixation behavior for multiple object tracking.

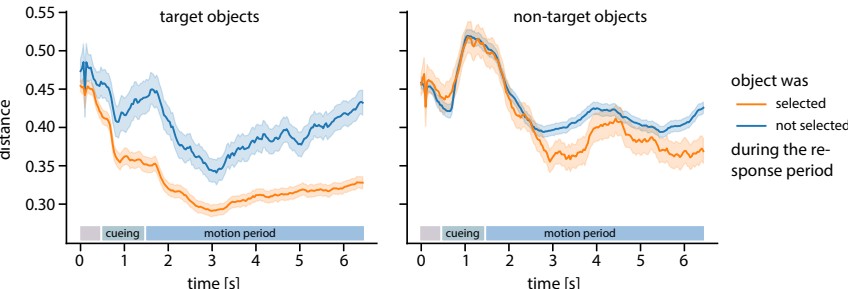

Figure 5: **Fixation behavior is related to the tracking task**. Average distance of target (left) and non-target objects (right) to the fixation and as a function of whether the object was believed to be a target object as indicated by the behavioral response. Distance as a fraction of the vertical (horizontal) extent of the motion area (i.e., relative to a motion area of $1 \times 1$).

