# OpenReview forum: "Human-like multiple object tracking through occlusion via gaze-following"
_NeurIPS.cc/2023/Workshop/Gaze_Meets_ML — Gaze Meets ML 2023 Poster_

### Official Review · Reviewer_Kw2w · 2023-10-23
**Human-like multiple object tracking through occlusion via gaze-following**

**Rating:** 9
**Confidence:** 5

**Review:**

This study sought to determine if human gaze behaviour can help uncover the differences between humans and model behaviour. The authors justified why the MOT model is the most suitable model for their study, but the author did not state whether bias was introduced into the training model. In terms of quality and clarity, the result of this study seems promising, and the figures were nicely presented. In terms of originality, it would be interesting if the authors compared their model to existing models to estimate its performance. Overall, this study reinstates the significance of gaze methods.

---

### Official Review · Reviewer_BPjv · 2023-10-24
**Relevant and useful findings that should influence the field of machine vision positively**

**Rating:** 9
**Confidence:** 2

**Review:**

Quality: The paper was well researched, well written, detailed, and relevant to future work that could address current obstacles in the field of machine vision and any tools that are developed in that field.
Clarity: A well detailed Appendix helped address any questions I had about the approach and strategy of the research. For instance, the appendix explained the process of fixation and fovea which the researchers took.
Significance: As mentioned above, this work can be very relevant in developing future tools added by machine vision and tracking. For instance, any future work that aims to support accessibility can certainly benefit from the findings of this paper.

---

### Official Review · Reviewer_fyzx · 2023-10-24
**Human-like multiple object tracking through occlusion via gaze-following**

**Rating:** 5
**Confidence:** 4

**Review:**

Strengths:
(+) Rigorous adaption of human vision to MOT
(+) Evaluation against human data (N = 9) that demonstrates ability to replicate human behavior

Weaknesses:
(-) No formal related works section
(-) The motivation and application of replicating human behavior in this context is not clear
(-) Lack of discussion related to the no-noise model performing best and what this means for other domains or research progress in improving MOT models by leveraging human behavior.

The paper presents a method built on Kalman Filtering and human vision to modify existing slot-based MOT models to replicate human behavior. Specifically, it is motivated that humans are good at tracking a reasonable set of moving objects in the presence of occlusions, while models are extremely good at at tracking an intractable set of objects but fail when they are occluded. The paper presents a slot-based model tracking extension that factors in characteristics of human vision.

The strengths of the paper is well written and easy to follow, with clearly presented results and intuitive explanations. The evidence presented supports the authors claim that the presented model extension is able to more closely replicate human behavior on the same MOT task.

However, the weaknesses of the paper then must be considered when evaluating the research contribution. While MOT is motivated and a strong description of slot-based MOT models are provided, the main takeaway of how the modified model would be used are not present. Specifically, in this case the unmodified model is still performing best. The observation of MOT models underperforming in the presence of occlusions, the original motivation for leveraging human behavior, is not explictly discussed in this context. Currently, the paper demonstrates manners to match human behavior, but does not show a benefit to MOT on this task and dataset. The research contribution thus relies on the authors discussing this result, and discussion on why this result was found (8 objects may be too low) and what it means for the field. Currently, the results are discussed and a reasonable argument is made for why their model parameters perform closer to human behavior. The situation in which MOT benefits from the human-based model and future work to evaluate and prove there is a benefit are ommited, along with a formal related works section. The description of slot-based trackers is clear, but do other multiple-object tracker models exist? Do they see the same issues with occlusion as slot-based trackers? All of this information is necessary to contextualize the work in the field and within this workshop for the broader research and MOT practicioner community. As stated above, the reader doesn't know what to make of this particular result.

As someone familar with perceptual studies and leveraging perception models accurate to humans, I can see rich applications for being able to accurately model human vision and object tracking. Knowing not what the best offline model can do, but how a human likely would perform has applications in several domains, including perceptual optimizations in graphics, gaze prediction tasks, and evaluating UX designs and interactive experiences. Thus, I encourage the authors to continue this line of work which is rigorously strong, but does not have a clear motivation or interpretation in its current state. The paper may meet the contribution level of the workshop through text justifications, a brief but necessary related work, and a slightly expanded discussion and future work section. Escalation to a full paper could be achieved by evaluation and demonstrating the ability of the approach to model human vision and benefit a research application as well.

---

### Meta-Review · Area_Chair_1fUN · 2023-10-26

**Recommendation:** Accept (Poster)
**Confidence:** 3

**Metareview:**

This paper shows how human gaze behavior helps explain the differing performance of state-of-the-art multiple object tracking models and human observers, with models matching human behavior when gaze behavior is imposed.

While results on human similarity are interesting, and the adaption of human vision to multiple object tracking is rigorous and well explained, authors should include a more extensive discussion of related work and better explain the motivation of the current approach for the MOT.

---

### Decision · Program_Chairs · 2023-10-26

Accept (Poster)